# Performance evaluation of the Roche Elecsys® Anti-SARS-CoV-2 immunoassays by comparison with neutralizing antibodies and clinical assessment

Satomi Takei[1,2], Tomohiko Ai[2], Takamasa Yamamoto[1], Gene Igawa[1], Takayuki Kanno[3], Minoru Tobiume[3], Makoto Hiki[4,5], Kaori Saito[2], Abdullah Khasawneh[2], Mitsuru Wakita[1], Shigeki Misawa[1], Takashi Miida[2], Atsushi Okuzawa[6,7], Tadaki Suzuki[3], Kazuhisa Takahashi[6,8], Toshio Naito[6,9], Yoko Tabe[2,6]*

1 Department of Clinical Laboratory, Juntendo University Hospital, Tokyo, Japan, 2 Department of Clinical Laboratory Medicine, Juntendo University Graduate School of Medicine, Tokyo, Japan, 3 Department of Pathology, National Institute of Infectious Diseases, Tokyo, Japan, 4 Department of Emergency Medicine, Juntendo University Faculty of Medicine, Tokyo, Japan, 5 Department of Cardiovascular Biology and Medicine, Juntendo University Faculty of Medicine, Tokyo, Japan, 6 Department of Research Support Utilizing Bioresource Bank, Juntendo University Graduate School of Medicine, Tokyo, Japan, 7 Department of Coloproctological Surgery, Juntendo University Graduate School of Medicine, Tokyo, Japan, 8 Department of Respiratory Medicine, Juntendo University Graduate School of Medicine, Tokyo, Japan, 9 Department of General Medicine, Juntendo University Graduate School of Medicine, Tokyo, Japan

* tabe@juntendo.ac.jp

**Data Availability Statement:** All relevant data are within the paper.

## Abstract

Quantitative measurement of SARS-CoV-2 neutralizing antibodies is highly expected to evaluate immune status, vaccine response, and antiviral therapy. The Elecsys® Anti-SARS-CoV-2 S (Elecsys® anti-S) was developed to measure anti-SARS-CoV-2 S proteins. We sought to investigate whether Elecsys® anti-S can be used to predict neutralizing activities in patients' serums using an authentic virus neutralization assay. One hundred forty-six serum samples were obtained from 59 patients with COVID-19 at multiple time points. Of the 59 patients, 44 cases were included in Group M (mild 23, moderate 21) and produced 84 samples (mild 35, moderate 49), while 15 cases were included in Group S (severe 11, critical 4) and produced 62 samples (severe 43, critical 19). The neutralization assay detected 73% positive cases, and Elecsys® anti-S and Elecsys® Anti-SARS-CoV-2 (Elecsys® anti-N) showed 72% and 66% positive cases, respectively. A linear correlation between the Elecsys® anti-S assay and the neutralization assay were highly correlated ($r = 0.7253$, $r^2 = 0.5261$) than a linear correlation between the Elecsys® anti-N and neutralization assay ($r = 0.5824$, $r^2 = 0.3392$). The levels of Elecsys® anti-S antibody and neutralizing activities were significantly higher in Group S than in Group M after 6 weeks from onset of symptoms ($p < 0.05$). Conversely, the levels of Elecsys® anti-N were comparable in both groups. Three immunosuppressed patients, including cancer patients, showed low levels of anti-S and anti-N antibodies and neutralizing activities throughout the measurement period, indicating the need for careful follow-up. Our data indicate that Elecsys® anti-S can predict the neutralization antibodies in COVID-19.

**Funding:** This research was supported by AMED (JP20fk0108472 to TN) and by Japan Society for the Promotion of Science Grants-in Aid for Scientific Research (22K15675 to ST).

**Competing interests:** The reagents used in this study were provided by Roche, but the study was performed by scientifically proper methods without any bias. This does not alter our adherence to PLOS ONE policies on sharing data and materials.

## Introduction

Coronavirus disease 2019 (COVID-19), caused by SARS-CoV-2, has been mainly diagnosed by reverse transcription-polymerase chain reaction (RT-PCR) that can directly detect the viral genomes [1]. Antigen testing has been also developed to detect pathogens rapidly without complicated procedures [2]. However, these tests cannot detect SARS-CoV-2 in certain periods after infection [3]. On the other hand, serological tests are essential tools to evaluate neutralizing antibody titers upon vaccination and to assess SARS-CoV-2 seroprevalence in cohorts [4, 5]. Neutralizing antibodies targeting the receptor-binding domain (RBD) of the spike (S) protein can reduce viral infectivity by binding to the surface epitopes of viral particles which blocks virus entry [6]. Although the authentic virus neutralization assays can directly measure the neutralizing activities of the SARS-CoV-2 virus, those methods need to be performed in Biosafety Level 3 facilities, which limits their application [7]. Therefore, there is a need for safer, high-throughput, and widely available measurement methods that correlate well with neutralizing activities.

Recently, the Elecsys® Anti-SARS-CoV-2 S (Elecsys® anti-S) has been developed to quantitatively measure total antibodies against the S protein RBD, and the Elecsys® Anti-SARS-CoV-2 (Elecsys® anti-N) has been developed to semi-quantitatively measure total antibodies against SARS-CoV-2 N proteins that regulates viral replication [6] (Roche Diagnostics International Ltd, Rotkreuz, Switzerland). The aim of this study was to evaluate the feasibility and usefulness of these immunoassays by comparing the chronological seroprevalences in patients with various severity of COVID-19 along with the neutralizing activities measured by an authentic virus neutralization assay.

## Materials and methods

### Clinical characteristics

This study complied with all relevant national regulations and institutional policies and was conducted in accordance with the tenets of the Declaration of Helsinki and was approved by the Institutional Review Board (IRB) at Juntendo University Hospital (IRB # 20–036). The need for informed consent from individual patients was waived because all samples were de-identified in line with the Declaration of Helsinki. From April to August 2020, 146 serum samples were collected from 59 patients with symptomatic COVID-19 confirmed by RT-PCR at multiple time points (number of samples per patient, median 2, IQR [2, 3]). The periods between different time points were 2 to 58 days. Of the 59 COVID-19 patients, including 58 inpatients and one outpatient, 44 cases were included Group M (mild 23, moderate 21) and produced 84 samples (mild 35, moderate 49), while 15 cases were included Group S (severe 11, critical 4) and produced 62 samples (severe 43, critical 19). Clinical data were retrospectively collected from patients' charts. All samples were from the unvaccinated patients.

### Laboratory assays

**Elecsys® Anti-SARS-CoV-2 S and Elecsys® Anti-SARS-CoV-2 immunoassays.** Serum samples were tested with the automated serological immunoassays, Elecsys® Anti-SARS-CoV-2 S (Elecsys® anti-S, Cat # 0928926750) and the Elecsys® Anti-SARS-CoV-2 (Elecsys® anti-N, Cat # 09203095501), by detecting antibodies to the receptor-binding domain (RBD) of S protein and antibodies to the N protein of SARS-CoV-2 (Roche Diagnostics) [8]. These assays received emergency use authorization approval from the US Food and Drug Administration (https://www.fda.gov/media/137605 (2020)). All samples were processed according to the manufacturer's instructions.

The results by Elecsys® anti-S were quantitatively shown in units of U/mL with the cut-off point of 0.80 U/mL to differentiate samples as reactive ($\geq$ 0.80 U/mL) and non-reactive ($<$ 0.80 U/mL). The values between 0.40–250 U/mL represented a linear range, and the results below this range were set to 0.4 U/mL. The samples above 250 U/mL were diluted into the linear range of the assay (1:10 or 1:100) with Diluent Universal reagent (Roche Diagnostics, Rotkreuz, Switzerland). Thus, the applied setting enabled an upper limit of quantification of 25000 U/mL for these analyses before dilution. The Elecsys® anti-N is a semi-quantitative assay, and the results were interpreted as follows: cutoff index (COI) $<$1.0 was non-reactive, and $\geq$1.0 was reactive.

**Neutralization assay.** The SARS-CoV-2 ancestral strain, WK-521 (lineage A, GISAID ID: EPI_ISL_408667), was used for the authentic virus neutralization assay that has been performed at the National Institute of Infectious Diseases (NIID) with ethics approval by the medical research ethics committee of NIID for the use of human subjects (#1178). Authentic virus neutralization assay has been performed as described previously [7]. Briefly, serially diluted serum samples (2-fold serial dilutions starting at 1:5 dilution, diluted with high glucose Dulbecco's Modified Eagle Medium supplemented with 2% Fetal Bovine Serum and 100 U/mL penicillin/streptomycin, from Fujifilm Wako Pure Chemicals, Japan) were mixed with the virus from 100 Median Tissue Culture Infectious Dose (TCID$_{50}$) and incubated at 37˚C for 1 hour. The mixture was subsequently incubated with VeroE6/TMPRSS2 cells (JCRB1819, JCRB Cell Bank, Japan) and seeded in 96-well flat-bottom plates for 4–6 days at 37˚C in a chamber supplied with 5% CO$_2$. Then the cells were fixed with 20% formalin (Fujifilm Wako Pure Chemicals) and stained with crystal violet solution (Sigma-Aldrich, St Louis, MO). Each sample was assayed in 2–4 wells and the average cut-off dilution index of $>$50% cytopathic effect was presented as a neutralizing titer. Neutralizing titer of the sample below the detection limit (1:5 dilution) was set as 2.5. Neutralizing antibody titer of $<$5 is considered negative and $>$5 is considered positive.

### Statistical analysis

Correlation studies were performed using Spearman's coefficient. Assay performance, linear regression, and curve fitting calculations were performed using Prism 9 (GraphPad Software, LLC, San Diego, CA, USA). For experiments involving two group comparisons, Wilcoxon signed-rank test was performed. The following notation was used to show statistical significance: * $p$ value $<$0.05, ** $p$ value $<$0.01, and *** $p$ value $<$0.001.

### Results

As shown in Table 1, patients were classified into two groups according to the WHO criteria [WHO. Clinical management of COVID-19. Available from: https://www.who.int/publications/i/item/clinical-management-of-covid-19]: Group M included mild and moderate cases and Group S included severe and critical cases.

We first compared the results of Elecsys® anti-S and Elecsys® anti-N to those of the authentic virus neutralizing assay. Of the 146 samples, the neutralization assay detected 73% (106/146) positives, and Elecsys® anti-S and Elecsys® anti-N showed 72% (105/146) and 66% (97/146) positives, respectively. These results were plotted, and the positive results were fitted with a linear regression. Fig 1A shows a linear correlation between the Elecsys® anti-S assay and the neutralization assay ($r$ = 0.7253, $r^2$ = 0.5261), and Fig 1B shows a linear correlation between the Elecsys® anti-N and neutralization assay ($r$ = 0.5824, $r^2$ = 0.3392).

We next examined the neutralizing activities (Fig 2A) and levels of antibodies (Fig 2B and 2C) at various time points after onset of symptoms. Fig 2 shows that the levels of neutralizing

**Table 1. Clinical characteristics of patients with COVID-19.**

| Disease severity[a] | Group M | | Group S | |
|---|---|---|---|---|
| | **Mild** | **Moderate** | **Severe** | **Critical** |
| Patient number (n = 59) | 39% (23/59) | 36% (21/59) | 19% (11/59) | 7% (4/59) |
| Male, % | 78% (18/23) | 57% (12/21) | 100% (11/11) | 75% (3/4) |
| Age range (average) | 24–82 (43.3) | 18–80 (54.9) | 6–86 (66.5) | 67–79 (75.3) |
| Past medical history | | | | |
| Hypertension | 9% (2/23) | 14% (3/21) | 9% (1/11) | 50% (2/4) |
| Hyperlipidemia | 9% (2/23) | 5% (1/21) | 0% (0/11) | 0% (0/4) |
| Diabetes | 0% (0/23) | 14% (3/21) | 18% (2/11) | 25% (1/4) |
| Cancer | 9% (2/23) | 0% (0/21) | 18% (2/11) | 25% (1/4) |
| Renal failure | 0% (0/23) | 0% (0/21) | 9% (1/11) | 25% (1/4) |
| Others, None known | 83% (19/23) | 76% (16/21) | 64% (7/11) | 50% (2/4) |
| Sample percentage (n = 146) | 24% (35/146) | 34% (49/146) | 29% (43/146) | 13% (19/146) |
| Days from onset | | | | |
| 0–6 (n = 25) | 7 | 8 | 8 | 2 |
| 7–13 (n = 28) | 7 | 11 | 10 | 0 |
| 14–20 (n = 26) | 9 | 8 | 7 | 2 |
| 21–27 (n = 24) | 4 | 8 | 6 | 6 |
| 28–34 (n = 15) | 3 | 5 | 5 | 2 |
| 35–41 (n = 13) | 4 | 3 | 4 | 2 |
| >42 (n = 15) | 1 | 6 | 3 | 5 |

aWHO criteria.

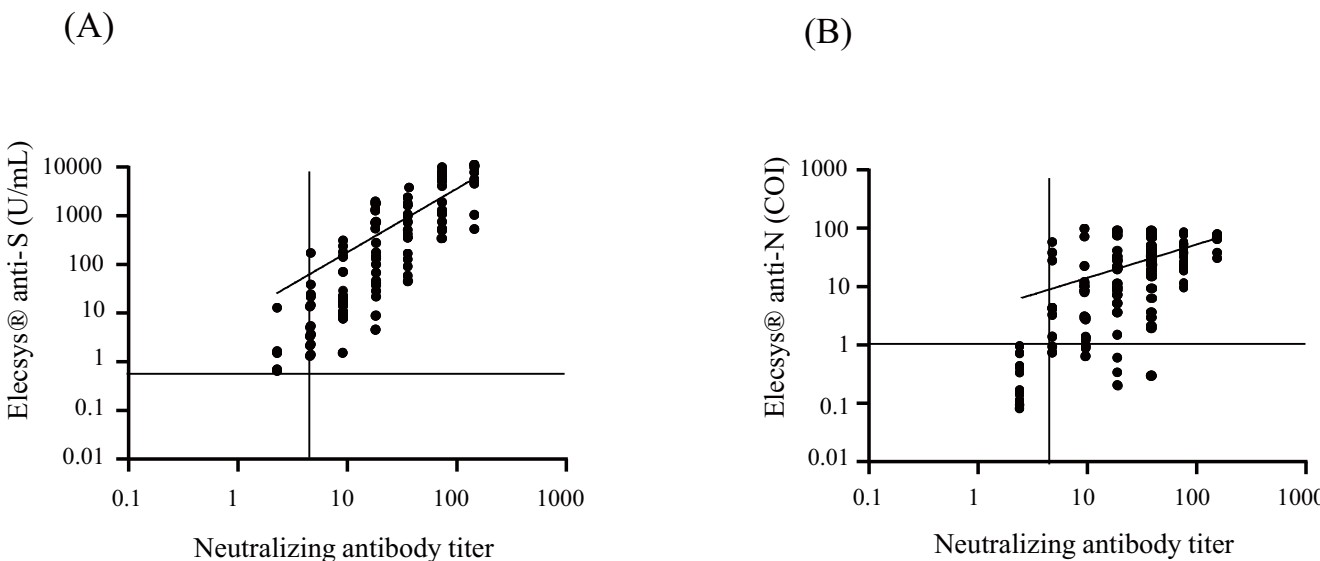

**Fig 1. Correlations of Elecsys® Anti-SARS-CoV-2 S and Elecsys®Anti-SARS-CoV-2 assays results to neutralization assay.** One hundred forty-six serum samples from COVID-19 patients were tested by Elecsys® anti-S, Elecsys® anti-N, and neutralization assay and were examined for correlations. (A) Correlation of Elecsys® anti-S and neutralization assay. (B)Correlation of Elecsys® anti-N and neutralization assay. Dotted lines represent the manufacturer's positive cutoff values: Elecsys® anti-S, 0.8 U/ml; Elecsys® anti-N, COI 1.0; neutralization assay, titer 5. The horizontal axis and the vertical axis are in logarithmic notations. Correlation studies were performed using Spearman's coefficient.

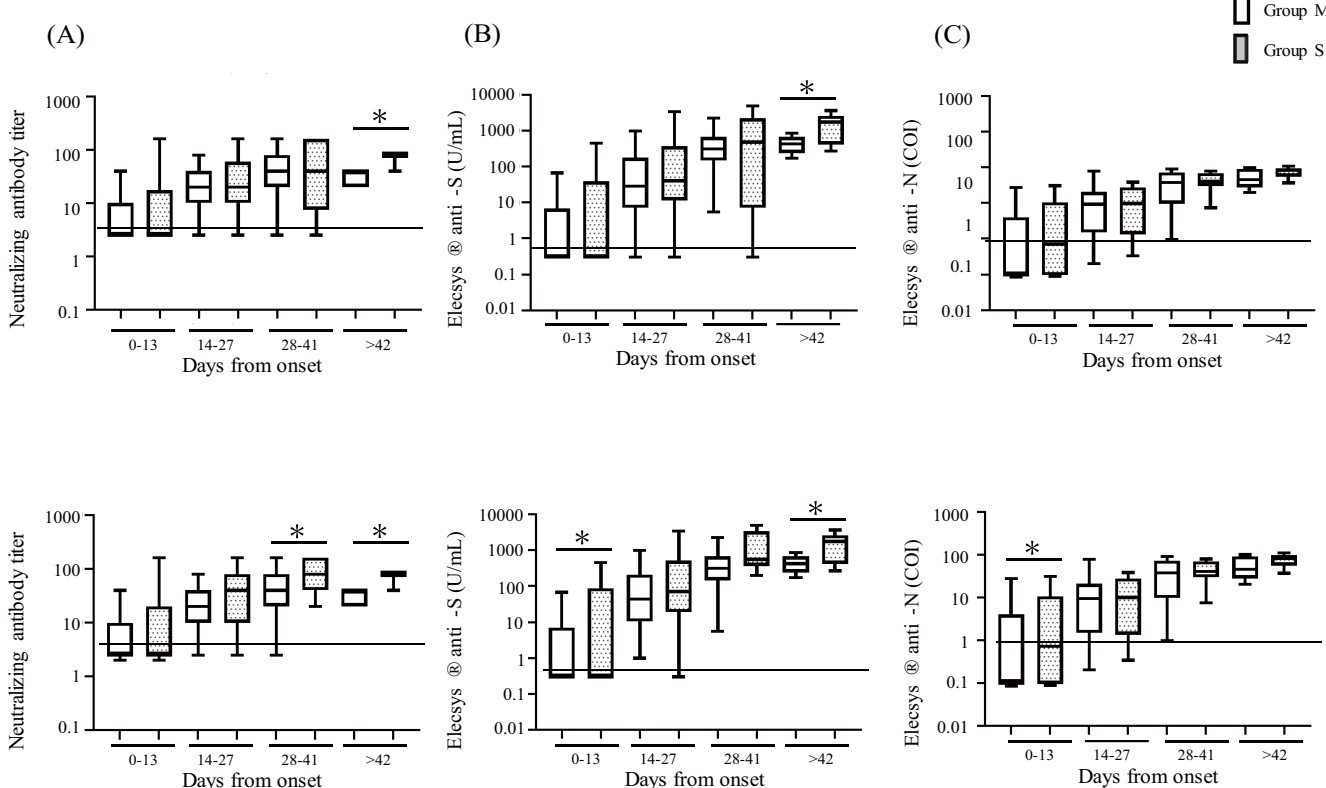

**Fig 2. Comparison of antibody levels between Group M (mild + moderate) and Group S (severe + critical).** One hundred forty-six serum samples from COVID-19 patients at various time points after the onset of symptoms were tested by neutralization assay, Elecsys® anti-S, and Elecsys® anti-N, and were examined for changes over time. All cases of antibody values of neutralization assay (A), Elecsys® anti-S (B), and Elecsys® anti-N (C) in Group M (n = 84) and S (n = 62) subjects were shown. Four cases receiving immunosuppressive therapy (1 in Group M; 3 in Group S) were excluded and the antibody values of neutralization assay (D), Elecsys® anti-S (E), and Elecsys® anti-N (F) were shown. Open bars represent Group M and dotted bars represent Group S. Wilcoxon signed-rank test was performed. The data were presented as median with interquartile ranges (IQR). *$p < 0.05$. The vertical axes are in logarithmic notation.

activities and antibodies tended to increase over time. However, there was no significant difference between Groups M and S until the sixth week. After the seventh week, the neutralization assay and Elecsys® anti-S showed significantly higher values in Group S than in Group M ($p < 0.05$), which was not observed in the Elecsys® anti-N results. Table 2 summarizes the details.

Since immunosuppressive therapies may cause false-negative results in antibody tests [9], the results after removing three patients receiving immunosuppressive therapies are shown in Fig 2D–2F. Fig 2D shows that the levels of neutralizing activities were higher in Group S than M after the fourth week ($p < 0.05$). Fig 2E shows that the levels of anti-S antibodies were significantly higher in Group S compared to Group M within the first 2 weeks and after the seventh week ($p < 0.05$). Fig 2F shows that the levels of anti-N antibodies were higher in Group S than M only in the first 2 weeks ($p < 0.05$).

Chronological changes in the results of the neutralization assay, Elecsys® anti-S assay, and Elecsys® anti-N assay were examined simultaneously in 23 inpatients who were tested in three time points or more. Table 3 summarizes the clinical characteristics. Fig 3 shows the line plots of the results in the patients of Group M. One patient (Pt #1) on immunosuppressive treatment for rheumatoid arthritis showed suppressed antibody responses for all three tests. The other patients' results became positive between 10 and 39 days from symptom onset.

**Table 2. Time course of seroprevalence by neutralization assay, Elecsys® Anti-SARS-CoV-2 S, and Elecsys® Anti-SARS-CoV-2.**

| Days from onset | sample number | Group M n = 84 (n = 80) | | | | | | | | Group S n = 62 (n = 47) | | | | | | | | |
|---|---|---|---|---|---|---|---|---|---|---|---|---|---|---|---|---|---|---|
| | | Virus neutralization assay | | Elecsys® Anti-SARS-CoV-2 S (Elecsys® anti-S) | | Elecsys® Anti-SARS-CoV-2 (Elecsys® anti-N) | | | | sample number | Virus neutralization assay | | Elecsys® Anti-SARS-CoV-2 S (Elecsys® anti-S) | | Elecsys® Anti-SARS-CoV-2 (Elecsys® anti-N) | | |
| | | Positive | % | Positive | % | Positive | % | | | | Positive | % | Positive | % | Positive | % |
| 0–6 | 15 (14) | 3 (3) | 20% (21%) | 3 (3) | 20% (21%) | 2(2) | 13% (14%) | | | 10 (7) | 1 (1) | 10% (14%) | 1 (1) | 10% (14%) | 1 (1) | 10% (14%) |
| 7–13 | 18 (17) | 11 (11) | 61% (65%) | 10 (10) | 56% (59%) | 9 (9) | 50% (53%) | | | 10 (6) | 5 (5) | 50% (83%) | 5 (5) | 50% (83%) | 5 (5) | 50% (83%) |
| 14–20 | 17 (16) | 16 (16) | 94% (100%) | 16 (16) | 94% (100%) | 11 (11) | 65% (69%) | | | 9 (7) | 8 (7) | 89% (100%) | 7 (7) | 78% (100%) | 8 (7) | 89% (100%) |
| 21–27 | 12 (11) | 11 (11) | 92% (100%) | 11 (11) | 92% (100%) | 11 (11) | 92% (100%) | | | 12 (10) | 11 (10) | 92% (100%) | 11 (10) | 92% (100%) | 10 (9) | 83% (90%) |
| 28–34 | 8 (8) | 8 (8) | 100% (100%) | 8 (8) | 100% (100%) | 8 (8) | 100% (100%) | | | 7 (5) | 6 (5) | 86% (100%) | 6 (5) | 86% (100%) | 6 (5) | 86% (100%) |
| 35–41 | 7 (7) | 7 (7) | 100% (100%) | 7 (7) | 100% (100%) | 7 (7) | 100% (100%) | | | 6 (4) | 5 (4) | 83% (100%) | 5 (4) | 83% (100%) | 5 (4) | 83% (100%) |
| >42 | 7 (7) | 7 (7) | 100% (100%) | 7 (7) | 100% (100%) | 7 (7) | 100% (100%) | | | 8 (8) | 8 (8) | 100% (100%) | 8 (8) | 100% (100%) | 8 (8) | 100% (100%) |

Numbers in parenthesis indicate the sample numbers after removing the samples from 3 patients under immunosuppressive therapy.

Fig 4 shows the line plots of the results in Group S. Three patients were treated with plasmapheresis. Regardless of the timing of plasmapheresis, one patient (Pt #21) remained negative, Pt #22 kept relatively low values, and Pt #23 showed high values in all assays. We observed no significant decrease of antibody levels of Pt #15 and Pt #23 which were continuously measured until 69 and 58 days after the onset of symptoms, respectively.

## Discussion

In this study, we demonstrated that the Elecsys® anti-S assay detected anti-spike (S) protein RBD antibodies at about the same time that the virus neutralization assay detected neutralizing activity: 2 weeks after onset of symptoms. These results are consistent with a recent analysis of naturally acquired SARS-CoV-2 infected individuals showing that neutralizing antibodies were almost detectable about 14 days after infection [10]. A previous study has further reported that production of neutralizing antibodies within 14 days of onset of symptoms is an important factor in recovery [11]. In this study, we detected that there is a humoral increase in S-specific antibodies with neutralizing activity. The measurement values of Elecsys® anti-S and neutralization assays were highly correlated, and the cases of Group S showed significantly higher values than Group M during the late phase of infection. The positive rate of the Elecsys® anti-N assay was lower than that of the Elecsys® anti-S and virus neutralization assays.

The measurement values of Elecsys® anti-S and neutralization assays were highly correlated, and the cases of Group S showed significantly higher values than Group M during the late phase of infection. When patients receiving immunosuppressive treatment were removed from the cohort, both the Elecsys® anti-S and Elecsys® anti-N assays detected significantly higher levels of antibody in Group S than in Group M in the early stages of infection. It has been shown that patients undergoing immunosuppressive treatment have higher incidence rates and serious adverse events of COVID-19 [12]. Antibody levels in patients receiving chemotherapy, radiation therapy, and other immunosuppressive therapies require careful

**Table 3. Clinical characteristics of inpatients with COVID-19.**

| Patient# | Disease severity[a] | Age (y) | Sex | Past medical history | Therapy | | Outcome |
|---|---|---|---|---|---|---|---|
| 1 | Mild | 65 | M | Hepatic cancer, Rheumatoid arthritis | N/A | N/A | cure/discharge |
| 2 | Mild | 63 | M | None known | N/A | N/A | cure/discharge |
| 3 | Mild | 72 | M | None known | N/A | Favipiravir, Ciclesonide, DEX | cure/discharge |
| 4 | Mild | 82 | M | Bile duct cancer | N/A | Favipiravir | cure/discharge |
| 5 | Moderate | 64 | M | post-Pancreatic Cancer | N/A | N/A | cure/discharge |
| 6 | Moderate | 78 | M | None known | N/A | Ciclesonide, Favipiravir | cure/discharge |
| 7 | Moderate | 41 | M | Lung sarcoidosis | N/A | Ciclesonide, Favipiravir | cure/discharge |
| 8 | Moderate | 76 | M | Prostatic hypertrophy | N/A | Ciclesonide, Favipiravir, Heparin | cure/discharge |
| 9 | Moderate | 37 | M | None known | N/A | Ciclesonide, Favipiravir | cure/discharge |
| 10 | Moderate | 59 | M | Hypertension, Hyperlipidemia | N/A | Ciclesonide, Favipiravir | cure/discharge |
| 11 | Moderate | 71 | F | Hyperlipidemia, Diabetes | N/A | Ciclesonide, Favipiravir, Heparin | cure/discharge |
| 12 | Moderate | 75 | F | Hypertension, Hyperlipidemia, Angina | N/A | Ciclesonide, Favipiravir | cure/discharge |
| 13 | Moderate | 18 | F | None known | N/A | N/A | cure/discharge |
| 14 | Moderate | 80 | F | Hypertension | N/A | Heparin | cure/discharge |
| 15 | Severe | 78 | M | Diabetes, Rheumatoid arthritis | O2 | Heparin | cure/discharge |
| 16 | Severe | 57 | M | Hyponatremia | O2 | Ciclesonide, Favipiravir | cure/discharge |
| 17 | Severe | 64 | M | Urinary stone | N/A | N/A | cure/discharge |
| 18 | Severe | 67 | M | Fatty liver, Kidney stones | O2 | Ciclesonide, Favipiravir, Heparin | cure/discharge |
| 19 | Severe | 46 | M | Diabetes, Angina, Stiff-person syndrome | O2 | Favipiravir, Heparin, mPSL, PSL, Remdesivir | cure/discharge |
| 20 | Severe | 84 | M | Colon cancer, Parkinson, Dementia | O2 | Heparin | cure/discharge |
| 21 | Severe | 84 | M | Hypertension, Lung cancer, Renal failure | O2 | CHDF, Ciclesonide, DEX, FFP, Favipiravir, mPSL, Plasmapheresis, PSL | death |
| 22 | Critical | 67 | M | Hypertension, Renal failure | Ventilation | CHDF, Ciclesonide, FFP, mPSL, Plasmapheresis | death |
| 23 | Critical | 77 | M | Hypertension, Diabetes, Prostate cancer | Ventilation | CHDF, Favipiravir, FFP, Heparin, mPSL, Plasmapheresis, PSL, Remdesivir | death |

Patients tested 3 times or more were included.

aWHO criteria.

Abbreviations: CHDF, Continuous hemodiafiltration; DEX, Dexamethasone; FFP, fresh frozen plasma; O2, Oxygen inhalation; PSL, prednisolone; N/A, not applicable.

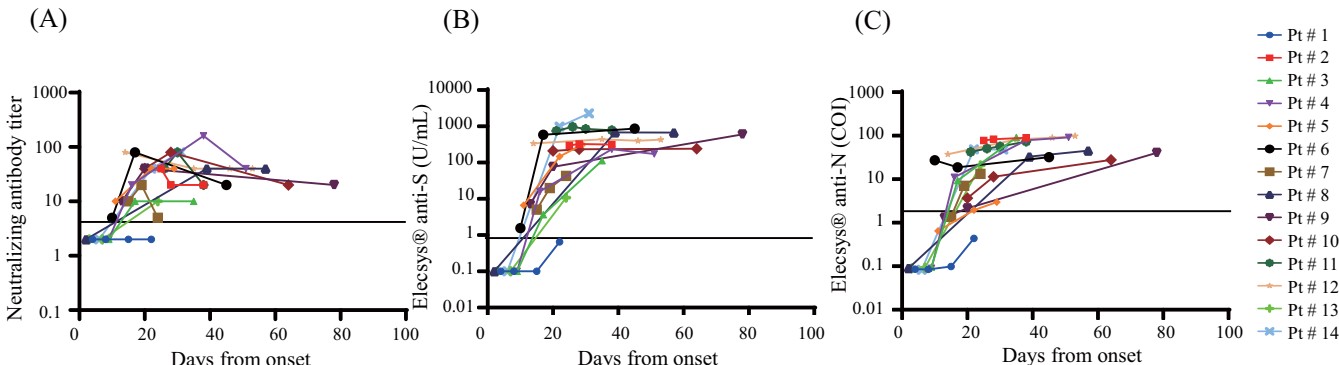

**Fig 3. Longitudinal change of antibodies in Group M.** Levels of SARS-CoV-2 antibodies in 14 mild to moderate cases (Group M) were tested. (A) Neutralizing antibody (B) Elecsys® anti-S (C) Elecsys® anti-N. The vertical axes are in logarithmic notation.

evaluation. It has been shown that IgG levels of SARS-CoV-2 S and N antibodies begin to decrease 2–3 months after infection [13, 14] and neutralizing titers begin to decrease 8 months later [15–17]. Regarding the effects of plasmapheresis treatment, a previous case report demonstrated that S-protein IgG increased after plasma exchange within 3 days, and moderately decreased from day 3 to day 7, without any change in N-protein IgG [18]. In this study, however, no significant changes in antibody levels caused by plasmapheresis were observed.

This study has several limitations: first, this study is a single-center study with limited sample size and short-term detections. Our results should be confirmed by additional assessments at other study sites. Second, variant determination has not been performed in the study. In terms of the SARS-CoV-2 strain that was used in the authentic virus neutralization assay, the utilized strain might have been different from the emerging variants. In Japan, lineage A and B variants were prevalent during the study period from April to August 2020 (1st and 2nd waves) [19]. We utilized the SARS-CoV-2 ancestral strain of lineage A for the authentic neutralization assay. On the other hand, the differences between the neutralizing antibody titers measured by the authentic neutralization assay and the commercially available immunoassays might depend on the viral variant, which is required to be verified in the future. Third, a comparably small sample size was utilized to measure agreement between the Elecsys® anti-N immunoassays and a neutralization assay; as such, these data would benefit from further evaluation and validation. Forth, this study does not include samples from asymptomatic patients

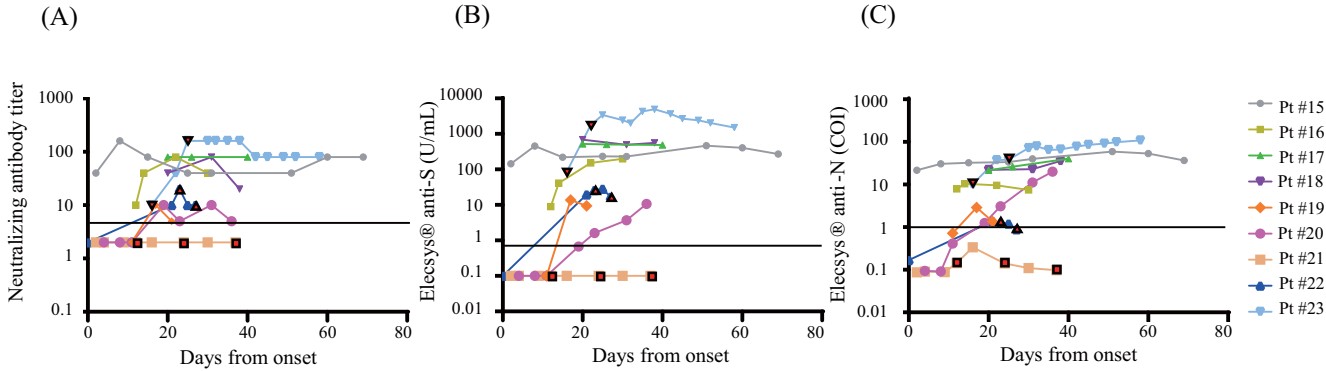

**Fig 4. Longitudinal change of antibodies in Group S.** Levels of SARS-CoV-2 antibodies in 9 severe to critical cases (Group S) were tested. (A) Neutralizing antibody (B) Elecsys® anti-S (C) Elecsys® anti-N. Colored boxes represent the days of plasmapheresis. The vertical axes are in logarithmic notation.

nor from vaccinated patients. Therefore, the validity of our findings for patients with asymptomatic/mildly symptomatic SARS-CoV-2 infection after vaccination is yet to be shown and requires further study.

In conclusion, we found good correlation between SARS-CoV-2 viral S protein RBD levels measured by Elecsys® anti-S and neutralizing antibody titers detected by an authentic virus neutralization assay. The findings of this study indicate that the quantitative detection of anti-SARS-CoV2 S protein by Elecsys® anti-S assay reliably quantifies the antibody response to SARS-CoV-2, which is highly relevant to estimate protection.

## Acknowledgments

The authors thank the Department of Research Support Utilizing Bioresource Bank, Juntendo University Graduate School of Medicine, for use of their facilities.

## Author Contributions

**Conceptualization:** Satomi Takei, Mitsuru Wakita, Shigeki Misawa, Yoko Tabe.

**Data curation:** Satomi Takei, Tomohiko Ai, Takamasa Yamamoto, Gene Igawa, Takayuki Kanno, Minoru Tobiume, Makoto Hiki, Abdullah Khasawneh, Tadaki Suzuki, Yoko Tabe.

**Formal analysis:** Satomi Takei, Tomohiko Ai, Takayuki Kanno, Minoru Tobiume, Kaori Saito, Takashi Miida, Atsushi Okuzawa, Tadaki Suzuki, Kazuhisa Takahashi, Toshio Naito.

**Funding acquisition:** Toshio Naito, Yoko Tabe.

**Investigation:** Satomi Takei, Makoto Hiki, Yoko Tabe.

**Methodology:** Takamasa Yamamoto, Gene Igawa, Kaori Saito.

**Project administration:** Mitsuru Wakita, Shigeki Misawa.

**Resources:** Makoto Hiki.

**Supervision:** Takashi Miida, Kazuhisa Takahashi.

**Validation:** Takashi Miida, Atsushi Okuzawa, Toshio Naito.

**Writing – original draft:** Satomi Takei, Yoko Tabe.

**Writing – review & editing:** Tomohiko Ai, Takayuki Kanno, Abdullah Khasawneh, Toshio Naito, Yoko Tabe.

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
