## [Editor Report · Decision Letter 0]

21 Mar 2022

PONE-D-22-05585Performance evaluation of the Roche Elecsys Anti-SARS-CoV-2 immunoassays by comparison with neutralizing antibodies and clinical assessmentPLOS ONE

Dear Dr. Tabe

Thank you for submitting your manuscript to PLOS ONE. After careful consideration, we feel that it has merit but does not fully meet PLOS ONE’s publication criteria as it currently stands. Therefore, we invite you to submit a revised version of the manuscript that addresses the points raised during the review process. The table fonts are very small. The figures are hazy and blurry. The abstract should include detailed results with number. Please correct and resubmit

We look forward to receiving your revised manuscript.

Kind regards,

Gheyath K. Nasrallah

Academic Editor

PLOS ONE

Journal Requirements:

"The reagents used in this study were provided by Roche, but the study was performed by scientifically proper methods without any bias.  "
---

## [Author Response · Author response to Decision Letter 0]

23 Mar 2022

March 23, 2022

Ms. Agatha Macaraig

Straive Editorial Assistant

RE: Resubmission of the revised manuscript titled “Performance evaluation of the Roche Elecsys Anti-SARS-CoV-2 immunoassays by comparison with neutralizing antibodies and clinical assessment” by Takei S., et al.

Ms. Macaraig,

We answer the requirements of the journal requirements as follows;

Reply (Answer; A) to the comments and questions (Comment; C):

Journal Requirements:

C1. Please ensure that your manuscript meets PLOS ONE's style requirements, including those for file naming. The PLOS ONE style templates can be found at 

A1. We corrected the manuscript following PLOS ONE's style requirements, including those for file naming.

C2. We note that the grant information you provided in the ‘Funding Information’ and ‘Financial Disclosure’ sections do not match. 

A2. We stated the financial disclosure in the cover letter. 

C3. Thank you for stating the following in the Competing Interests section: 

"The reagents used in this study were provided by Roche, but the study was performed by scientifically proper methods without any bias. "

A3. We included our updated Competing Interests statement in the cover letter as follows; 

"The reagents used in this study were provided by Roche, but the study was performed by scientifically proper methods without any bias. This does not alter our adherence to PLOS ONE policies on sharing data and materials.” 

C4. In your Data Availability statement, you have not specified where the minimal data set underlying the results described in your manuscript can be found. PLOS defines a study's minimal data set as the underlying data used to reach the conclusions drawn in the manuscript and any additional data required to replicate the reported study findings in their entirety. All PLOS journals require that the minimal data set be made fully available. For more information about our data policy, please see http://journals.plos.org/plosone/s/data-availability.

A4. We corrected the Data Availability statement as follows; “All relevant data are within the paper.“ 

Sincerely,

Yoko Tabe, M.D., Ph.D.

Professor 

Department of Clinical Laboratory Medicine, 

Juntendo University, School of Medicine

---

## [Decision Letter · Decision Letter 1]

9 May 2022

PONE-D-22-05585R1Performance evaluation of the Roche Elecsys Anti-SARS-CoV-2 immunoassays by comparison with neutralizing antibodies and clinical assessmentPLOS ONE

Dear Dr. Tabe,

Thank you for submitting your manuscript to PLOS ONE. After careful consideration, we feel that it has merit but does not fully meet PLOS ONE’s publication criteria as it currently stands. Therefore, we invite you to submit a revised version of the manuscript that addresses the points raised during the review process. Bothe reviewers has serious concerns about the methodology. Further Rev2 believes that the manuscript is poorly written and organized. Please make sure that you address all of comments of the reviewers before you submit  a revised version.   

We look forward to receiving your revised manuscript.

Kind regards,

Gheyath K. Nasrallah

Academic Editor

PLOS ONE

Reviewers' comments:

Reviewer's Responses to Questions

**Comments to the Author**

1. If the authors have adequately addressed your comments raised in a previous round of review and you feel that this manuscript is now acceptable for publication, you may indicate that here to bypass the “Comments to the Author” section, enter your conflict of interest statement in the “Confidential to Editor” section, and submit your "Accept" recommendation.

Reviewer #1: (No Response)

Reviewer #2: (No Response)

2. Is the manuscript technically sound, and do the data support the conclusions?

Reviewer #1: Yes

Reviewer #2: Partly

3. Has the statistical analysis been performed appropriately and rigorously? 

Reviewer #1: Yes

Reviewer #2: No

4. Have the authors made all data underlying the findings in their manuscript fully available?

Reviewer #1: Yes

Reviewer #2: No

5. Is the manuscript presented in an intelligible fashion and written in standard English?

Reviewer #1: Yes

Reviewer #2: No

6. Review Comments to the Author

Reviewer #1: Takei et al. investigated the correlation between the neutralizing antibody titre measured by two commercially available immunoassays and a standard virus neutralization assay. The measurement values of Elecsys Anti-SARS-CoV-2 S and neutralization assays were highly correlated and significantly higher values has been measured in the group of severe and critically ill cases. The study is of interest as serological tests are essential tools to evaluate the neutralizing antibodies targeting the receptor binding domain of the SARS-CoV-2 spike protein.

The authors might consider some minor comments:

Section Materials and methods, Neutralization assay:

It would be interesting to read which strain of SARS-CoV-2 was used in the assay, as the neutralizing activity of the antibodies may be different against the original SARS-CoV-2 strain and the emerging variants of concern. There may be differences between the neutralizing antibody titer measured by the commercially available immunoassays and the protective immunity, depending on the viral variant.

Section Results, Table 3:

Is there any information available regarding the SARS-CoV-2 vaccination status of these patients? Or they were all immunologically naïve prior to the acute infection?

In addition to the diagnostic PCR assays, has a variant determination been performed in the study group?

Reviewer #2: In this study, the authors investigated whether measuring anti-SARS-CoV-2 Spike antibodies can predict neutralizing activities in patients’ sera. A commercial assay (Elecsys® Anti-SARS-CoV-2) was utilized for measuring anti-S and anti-N SARS-CoV-2 antibodies. This was then compared to neutralizing titers utilizing an authentic virus neutralization assay.

Major comments:

1- The article is poorly written. All sections need to be improved. There are missing info in the abstract and methods. The authors did not discuss similar studies nor compared their results to literature in their discussion.

2- A major concern in the study in the correlation analysis. The authors indicated that 146 serum samples were tested, with a positivity rate of >70% for both binding and neutralization assays. However, the correlation figures (Fig 1A and 1B) included only few data points (n=6). All data points (n=146) need to be plotted along with a properly drawn fitted line, in addition, r, and r2 values need to be both mentioned in order to assess the strength of the correlation.

3- The abstract needs to be improved. The numbers and percentages need to be added accordingly with each finding.

4- Patients sample details needs to be better described in the Methods section. There is no information on how many samples were collected per patient. What is the number of inpatients and outpatients? Also, what is the duration between different time-points?

5- Figures and legends are poorly presented. Please fix the following:

a.Figure 1 needs re-plotting after including all the results.

b.In figure 2, better to label the lower panels with other labels (E,F and G), better than referring to them as “the lower panel of panel A, B or C”

c.Figure 3 is not clear at all. It is better to use colors to differentiate between patients.

d.In figure 3, why did they choose patients who were tested more than 3 time points only? Patients with 3 time-points can be included in the analysis especially if there is a good time gap between each time-point.

e.All legends need to be expanded to clearly indicate the type of test, analysis, and sample numbers.

6- In line 260, it is mentioned: “no significant reduction in antibody levels tested was observed until 69 days after the onset of symptoms”. I believe this was observed in 2 patients only whose samples were available at that time-point. This is not enough to draw conclusions. The limitation of sample size need to be clearly mentioned in the discussion.

Minor comments:

1- In the abstract, change "levels of neutral antibodies in COVID-19" to "neutralization antibodies"

2- The catalog numbers of the used commercial assays need to be mentioned in the methods.

3- Kindly indicate the type/source of the real virus used in the neutralization assay.

4- In the methods section, it is better to change “Clinical Backgrounds” to “Clinical characteristics”.

5- Table 1. If age is represented as range, please add this to the label of the row “Age range (average)”

6- Whenever “Elecsys® Anti-SARS-CoV-2” is mentioned, it is better to add “anti-N”, so it does not confuse the reader. This applies to figures and tables too.

7- Table 1 and 3 have the same title. Kindly change the title of table 3 to “inpatients”.

7. PLOS authors have the option to publish the peer review history of their article (what does this mean?). If published, this will include your full peer review and any attached files.

Reviewer #1: No

Reviewer #2: **Yes: **Maria K. Smatti

---

## [Author Response · Author response to Decision Letter 1]

24 Jun 2022

Reply (Answer; A) to Reviewer’s comments and questions (Comment; C):

Reviewer 1

COMMENTS by Reviewer #1: 

Takei et al. investigated the correlation between the neutralizing antibody titer measured by two commercially available immunoassays and a standard virus neutralization assay. The measurement values of Elecsys Anti-SARS-CoV-2 S and neutralization assays were highly correlated and significantly higher values has been measured in the group of severe and critically ill cases. The study is of interest as serological tests are essential tools to evaluate the neutralizing antibodies targeting the receptor binding domain of the SARS-CoV-2 spike protein.

COMMENTS (continued):

The authors might consider some minor comments:

C1: Section Materials and methods, Neutralization assay:

It would be interesting to read which strain of SARS-CoV-2 was used in the assay, as the neutralizing activity of the antibodies may be different against the original SARS-CoV-2 strain and the emerging variants of concern. There may be differences between the neutralizing antibody titers measured by the commercially available immunoassays and the protective immunity, depending on the viral variant.

A1: Thank you for the valuable suggestions. We added the sentence in the Materials and Methods section and discussed about the differences between the neutralizing antibody titer and the protective immunity, which might depend on the viral variant (L120-123, L275-282).

C2: Is there any information available regarding the SARS-CoV-2 vaccination status of these patients? Or they were all immunologically naïve prior to the acute infection?

A2: This study targeted only pre-vaccinated patients, who were immunologically naive prior to infection. We added this information to the Materials and Methods section (L98).

C3: In addition to the diagnostic PCR assays, has a variant determination been performed in the study group?

A3: Unfortunately, no variant determination has been performed in the study group. We mentioned this as the study limitation and discussed the SARS-CoV-2 strain used in the neutralization assay and the SARS-CoV-2 variant prevalent in Japan during the study period (L275-282).

Reviewer #2: 

In this study, the authors investigated whether measuring anti-SARS-CoV-2 Spike antibodies can predict neutralizing activities in patients’ sera. A commercial assay (Elecsys® Anti-SARS-CoV-2) was utilized for measuring anti-S and anti-N SARS-CoV-2 antibodies. This was then compared to neutralizing titers utilizing an authentic virus neutralization assay.

C1: The article is poorly written. All sections need to be improved. There are missing info in the abstract and methods. The authors did not discuss similar studies nor compared their results to literature in their discussion.

A1: Following the reviewer’s suggestion, we added more details in the Abstract and Materials and Method section (L44-55, L91-98). We also discussed and compared our results to previous literatures (L248-253, L263-269).

C2: A major concern in the study in the correlation analysis. The authors indicated that 146 serum samples were tested, with a positivity rate of >70% for both binding and neutralization assays. However, the correlation figures (Fig 1A and 1B) included only few data points (n=6). All data points (n=146) need to be plotted along with a properly drawn fitted line, in addition, r, and r2 values need to be both mentioned in order to assess the strength of the correlation.

A2: One hundred and forty-six samples were included in the correlation analysis. Although some of the plotted sample dots appear to overlap in the graphs in Fig 1A and 1B, all of 146 samples are included in these figures.

We appreciate the reviewer’s comments about the r2 values. We added the r2 along with the r values (L51-52, L155-156).

C3: The abstract needs to be improved. The numbers and percentages need to be added accordingly with each finding.

A3: Following the reviewer’s suggestion, we added the numbers and percentages in the Abstract section (L44-55).

C4: Patients sample details needs to be better described in the Methods section. There is no information on how many samples were collected per patient. What is the number of inpatients and outpatients? Also, what is the duration between different time-points?

A4: We added the detailed information of sample numbers and duration in the Materials and Methods section (L91-97).

C5: Figures and legends are poorly presented. Please fix the following:

a.　Figure 1 needs re-plotting after including all the results.

b.　In figure 2, better to label the lower panels with other labels (E,F and G), better than referring to them as “the lower panel of panel A, B or C”

c.　Figure 3 is not clear at all. It is better to use colors to differentiate between patients.

d.　In figure 3, why did they choose patients who were tested more than 3 time points only? Patients with 3 time-points can be included in the analysis especially if there is a good time gap between each time-point.

e.　All legends need to be expanded to clearly indicate the type of test, analysis, and sample numbers.

A5: 

a. One hundred and forty-six samples were included in the correlation analysis. Although some of the plotted sample dots appear to overlap in the graphs in Fig 1A and 1B, all of 146 samples are included in these figures.

b. We labeled the lower panels with D, E, and F in Figure 2.

c. We used colors to differentiate between patients in Figure 3.

d. We included patients with 3 time-points in the analysis and corrected the sentence in Table 3 abbreviation (L218) and the Result section as follows: “Chronological changes in the results of the neutralization assay, Elecsys® anti-S assay, and Elecsys® anti-N assay were examined simultaneously in 23 inpatients who were tested in three time points or more.” (L203-205).

e. Following the reviewer’s suggestion, we added the type of test, analysis, and sample numbers in legends of Fig 1 (L164-170) and Fig 2 (L192-200). 

C6: In line 260, it is mentioned: “no significant reduction in antibody levels tested was observed until 69 days after the onset of symptoms”. I believe this was observed in 2 patients only whose samples were available at that time-point. This is not enough to draw conclusions. The limitation of sample size need to be clearly mentioned in the discussion.

A6 Following the reviewer’s suggestion, we deleted the sentence “no significant reduction in antibody levels tested was observed until 69 days after the onset of symptoms” and mentioned the limitation of this study regarding sample size (L274-275).

C7: In the abstract, change "levels of neutral antibodies in COVID-19" to "neutralization antibodies"

A7: Corrected (L58).

C8: The catalog numbers of the used commercial assays need to be mentioned in the Materials and Methods.

A8: We added the catalog numbers of the used commercial assays in the methods section (L102-106).

C9: Kindly indicate the type/source of the real virus used in the neutralization assay.

A9: Information of SARS-CoV-2 strain used in the neutralization assay was provided in the Materials and Methods section (L120-123).

C10: In the methods section, it is better to change “Clinical Backgrounds” to “Clinical characteristics”.

A10: Following the reviewer’s suggestion, we changed “Clinical Backgrounds” to “Clinical characteristics”　(L86).

C11: Table 1. If age is represented as range, please add this to the label of the row “Age range (average)”

A11: Corrected “Age, y (average)” to “Age range (average)” (Table 1).

C12: Whenever “Elecsys® Anti-SARS-CoV-2” is mentioned, it is better to add “anti-N”, so it does not confuse the reader. This applies to figures and tables too.

A12: Following the reviewer’s suggestion, we corrected “Elecsys® Anti-SARS-CoV-2-S” to “Elecsys ® Anti-SARS-CoV-2-S (Elecsys® anti-S)” and “Elecsys® Anti-SARS-CoV-2” to “Elecsys ® Anti-SARS-CoV-2 (Elecsys® anti-N)” in manuscripts, figures, and tables.

C13: Table 1 and 3 have the same title. Kindly change the title of table 3 to “inpatients”

A13: Thank you for your kind and detailed review. We corrected “patients” to “inpatients” in the title of Table 3.

---

## [Decision Letter · Decision Letter 2]

24 Aug 2022

Performance evaluation of the Roche Elecsys ® Anti-SARS-CoV-2 immunoassays by comparison with neutralizing antibodies and clinical assessment

PONE-D-22-05585R2

Dear Dr. Tabe,

We’re pleased to inform you that your manuscript has been judged scientifically suitable for publication and will be formally accepted for publication once it meets all outstanding technical requirements.

Kind regards,

Gheyath K. Nasrallah

Academic Editor

PLOS ONE

Additional Editor Comments (optional):

Reviewers' comments:

Reviewer's Responses to Questions

**Comments to the Author**

1. If the authors have adequately addressed your comments raised in a previous round of review and you feel that this manuscript is now acceptable for publication, you may indicate that here to bypass the “Comments to the Author” section, enter your conflict of interest statement in the “Confidential to Editor” section, and submit your "Accept" recommendation.

Reviewer #2: All comments have been addressed

2. Is the manuscript technically sound, and do the data support the conclusions?

Reviewer #2: Yes

3. Has the statistical analysis been performed appropriately and rigorously? 

Reviewer #2: Yes

4. Have the authors made all data underlying the findings in their manuscript fully available?

Reviewer #2: Yes

5. Is the manuscript presented in an intelligible fashion and written in standard English?

Reviewer #2: Yes

6. Review Comments to the Author

Reviewer #2: (No Response)

7. PLOS authors have the option to publish the peer review history of their article (what does this mean?). If published, this will include your full peer review and any attached files.

Reviewer #2: No

---

## [Editor Report · Acceptance letter]

2 Sep 2022

PONE-D-22-05585R2 

Performance evaluation of the Roche Elecsys® Anti-SARS-CoV-2 immunoassays by comparison with neutralizing antibodies and clinical assessment 

Dear Dr. Tabe:

I'm pleased to inform you that your manuscript has been deemed suitable for publication in PLOS ONE. Congratulations! Your manuscript is now with our production department. 

Kind regards, 

on behalf of

Dr. Gheyath K. Nasrallah 

Academic Editor

PLOS ONE